# Nonparametric Online Regression
# while Learning the Metric

**Ilja Kuzborskij**
EPFL
Switzerland
ilja.kuzborskij@gmail.com

**Nicolò Cesa-Bianchi**
Dipartimento di Informatica
Università degli Studi di Milano
Milano 20135, Italy
nicolo.cesa-bianchi@unimi.it

## Abstract

We study algorithms for online nonparametric regression that learn the directions along which the regression function is smoother. Our algorithm learns the Mahalanobis metric based on the gradient outer product matrix $\boldsymbol{G}$ of the regression function (automatically adapting to the effective rank of this matrix), while simultaneously bounding the regret —on the same data sequence— in terms of the spectrum of $\boldsymbol{G}$. As a preliminary step in our analysis, we extend a nonparametric online learning algorithm by Hazan and Megiddo enabling it to compete against functions whose Lipschitzness is measured with respect to an arbitrary Mahalanobis metric.

## 1 Introduction

An online learner is an agent interacting with an unknown and arbitrary environment over a sequence of rounds. At each round $t$, the learner observes a data point (or instance) $\boldsymbol{x}_t \in \mathcal{X} \subset \mathbb{R}^d$, outputs a prediction $\widehat{y}_t$ for the label $y_t \in \mathbb{R}$ associated with that instance, and incurs some loss $\ell_t(\widehat{y}_t)$, which in this paper is the square loss $(\widehat{y}_t - y_t)^2$. At the end of the round, the label $y_t$ is given to the learner, which he can use to reduce his loss in subsequent rounds. The performance of an online learner is typically measured using the regret. This is defined as the amount by which the learner's cumulative loss exceeds the cumulative loss (on the same sequence of instances and labels) of *any* function $f$ in a given reference class $\mathcal{F}$ of functions,

$$R_T(f) = \sum_{t=1}^{T} \left( \ell_t(\widehat{y}_t) - \ell_t\big(f(\boldsymbol{x}_t)\big) \right) \qquad \forall f \in \mathcal{F} . \tag{1}$$

Note that typical regret bounds apply to all $f \in \mathcal{F}$ and to all individual data sequences. However, the bounds are allowed to scale with parameters arising from the interplay between $f$ and the data sequence.

In order to capture complex environments, the reference class of functions should be large. In this work we focus on nonparametric classes $\mathcal{F}$, containing all differentiable functions that are smooth with respect to some metric on $\mathcal{X}$. Our approach builds on the simple and versatile algorithm for nonparametric online learning introduced in [6]. This algorithm has a bound on the regret $R_T(f)$ of order (ignoring logarithmic factors)

$$\left( 1 + \sqrt{\sum_{i=1}^{d} \|\partial_i f\|_\infty^2} \right) T^{\frac{d}{1+d}} \qquad \forall f \in \mathcal{F} . \tag{2}$$

Here $\|\partial_i f\|_\infty$ is the value of the partial derivative $\partial f(\boldsymbol{x})\big/\partial x_i$ maximized over $\boldsymbol{x} \in \mathcal{X}$. The square root term is the Lipschitz constant of $f$, measuring smoothness with respect to the Euclidean metric.

However, in some directions $f$ may be smoother than in others. Therefore, if we knew in advance the set of directions along which the best performing reference functions $f$ are smooth, we could use this information to control regret better. In this paper we extend the algorithm from [6] and make it adaptive to the Mahalanobis distance defined through an arbitrary positive definite matrix $M$ with spectrum $\{(\boldsymbol{u}_i, \lambda_i)\}_{i=1}^d$ and unit spectral radius ($\lambda_1 = 1$). We prove a bound on the regret $R_T(f)$ of order (ignoring logarithmic factors)

$$\left( \sqrt{\det_\kappa(\boldsymbol{M})} + \sqrt{\sum_{i=1}^d \frac{\|\nabla_{\boldsymbol{u}_i} f\|_\infty^2}{\lambda_i}} \right) T^{\frac{\rho_T}{1+\rho_T}} \qquad \forall f \in \mathcal{F} \ . \tag{3}$$

Here $\rho_T \leq d$ is, roughly, the number of eigenvalues of $M$ larger than a threshold shrinking polynomially in $T$, and $\det_\kappa(\boldsymbol{M}) \leq 1$ is the determinant of $M$ truncated at $\lambda_\kappa$ (with $\kappa \leq \rho_T$). The quantity $\|\nabla_{\boldsymbol{u}_i} f\|_\infty^2$ is defined like $\|\partial_i f\|_\infty$, but with the directional derivative $\nabla f(\boldsymbol{x})^\top \boldsymbol{u}$ instead of the partial derivative. When the spectrum of $M$ is light-tailed (so that $\rho_T \ll d$ and, simultaneously, $\det_\kappa(\boldsymbol{M}) \ll 1$), with the smaller eigenvalues $\lambda_i$ corresponding to eigenvectors in which $f$ is smoother (so that the ratios $\|\nabla_{\boldsymbol{u}_i} f\|_\infty^2 / \lambda_i$ remain controlled), then our bound improves on (2). On the other hand, when no preliminary knowledge about good $f$ is available, we may run the algorithm with $M$ equal to the identity matrix and recover exactly the bound (2).

Given that the regret can be improved by informed choices of $M$, it is natural to ask whether some kind of improvement is still possible when $M$ is learned online, from the same data sequence on which the regret is being measured. Of course, this question makes sense if the data tell us something about the smoothness of the $f$ against which we are measuring the regret. In the second part of the paper we implement this idea by considering a scenario where instances are drawn i.i.d. from some unknown distribution, labels are stochastically generated by some unknown regression function $f_0$, and we have no preliminary knowledge about the directions along which $f_0$ is smoother.

In this stochastic scenario, the expected gradient outer product matrix $\boldsymbol{G} = \mathbb{E}\left[\nabla f_0(\boldsymbol{X}) \nabla f_0(\boldsymbol{X})^\top\right]$ provides a natural choice for the matrix $M$ in our algorithm. Indeed, $\mathbb{E}\left[\left(\nabla f_0(\boldsymbol{X})^\top \boldsymbol{u}_i\right)^2\right] = \mu_i$ where $\boldsymbol{u}_1, \ldots, \boldsymbol{u}_d$ are the eigenvectors of $\boldsymbol{G}$ while $\mu_1, \ldots, \mu_d$ are the corresponding eigenvalues. Thus, eigenvectors $\boldsymbol{u}_1, \ldots \boldsymbol{u}_d$ capture the principal directions of variation for $f$. In fact, assuming that the labels obey a statistical model $Y = g(\boldsymbol{B}\boldsymbol{X}) + \varepsilon$ where $\varepsilon$ is the noise and $\boldsymbol{B} \in \mathbb{R}^{k \times d}$ projects $\boldsymbol{X}$ onto a $k$-dimensional subspace of $\mathcal{X}$, one can show [21] that $\mathrm{span}(\boldsymbol{B}) \equiv \mathrm{span}(\boldsymbol{u}_1, \ldots, \boldsymbol{u}_d)$. In this sense, $\boldsymbol{G}$ is the "best" metric, because it recovers the $k$-dimensional relevant subspace.

When $\boldsymbol{G}$ is unknown, we run our algorithm in phases using a recently proposed estimator $\widehat{\boldsymbol{G}}$ of $\boldsymbol{G}$. The estimator is trained on the same data sequence and is fed to the algorithm at the beginning of each phase. Under mild assumptions on $f_0$, the noise in the labels, and the instance distribution, we prove a high probability bound on the regret $R_T(f_0)$ of order (ignoring logarithmic factors)

$$\left( 1 + \sqrt{\sum_{j=1}^d \frac{\left(\|\nabla_{\boldsymbol{u}_j} f_0\|_\infty + \|\nabla_V f_0\|_\infty\right)^2}{\mu_j / \mu_1}} \right) T^{\frac{\widetilde{\rho}_T}{1+\widetilde{\rho}_T}} \ . \tag{4}$$

Observe that the rate at which the regret grows is the same as the one in (3), though now the effective dimension parameter $\widetilde{\rho}_T$ is larger than $\rho_T$ by an amount related to the rate of convergence of the eigenvalues of $\widehat{\boldsymbol{G}}$ to those of $\boldsymbol{G}$. The square root term is also similar to (3), but for the extra quantity $\|\nabla_V f_0\|_\infty$, which accounts for the error in approximating the eigenvectors of $\boldsymbol{G}$. More precisely, $\|\nabla_V f_0\|_\infty$ is $\|\nabla_{\boldsymbol{v}} f\|_\infty$ maximized over directions $\boldsymbol{v}$ in the span of $V$, where $V$ contains those eigenvectors of $\boldsymbol{G}$ that cannot be identified because their eigenvalues are too close to each other (we come back to this issue shortly). Finally, we lose the dependence on the truncated determinant, which is replaced here by its trivial upper bound 1.

The proof of (2) in [6] is based on the sequential construction of a sphere packing of $\mathcal{X}$, where the spheres are centered on adaptively chosen instances $\boldsymbol{x}_t$, and have radii shrinking polynomially with time. Each sphere hosts an online learner, and each new incoming instance is predicted using the learner hosted in the nearest sphere. Our variant of that algorithm uses an ellipsoid packing, and computes distances using the Mahalanobis distance $\|\cdot\|_{\boldsymbol{M}}$. The main new ingredient in the analysis leading to (3) is our notion of effective dimension $\rho_T$ (we call it the *effective rank* of $M$), which measures how fast the spectrum of $M$ vanishes. The proof also uses an ellipsoid packing bound and a lemma relating the Lipschitz constant to the Mahalanobis distance.

The proof of (4) is more intricate because $G$ is only known up to a certain approximation. We use an estimator $\widehat{G}$, recently proposed in [14], which is consistent under mild distributional assumptions when $f_0$ is continuously differentiable. The first source of difficulty is adjusting the notion of effective rank (which the algorithm needs to compute) to compensate for the uncertainty in the knowledge of the eigenvalues of $G$. A further problematic issue arises because we want to measure the smoothness of $f_0$ along the eigendirections of $G$, and so we need to control the convergence of the eigenvectors, given that $\widehat{G}$ converges to $G$ in spectral norm. However, when two eigenvalues of $G$ are close, then the corresponding eigenvectors in the estimated matrix $\widehat{G}$ are strongly affected by the stochastic perturbation (a phenomenon known as hybridization or spectral leaking in matrix perturbation theory, see [1, Section 2]). Hence, in our analysis we need to separate out the eigenvectors that correspond to well spaced eigenvalues from the others. This lack of discrimination causes the appearance in the regret of the extra term $\left\| \nabla_V f_0 \right\|_\infty$.

## 2 Related works

Nonparametric estimation problems have been a long-standing topic of study in statistics, where one is concerned with the recovery of an optimal function from a rich class under appropriate probabilistic assumptions. In online learning, the nonparametric approach was investigated in [15, 16, 17] by Vovk, who considered regression problems in large spaces and proved bounds on the regret. Minimax rates for the regret were later derived in [13] using a non-constructive approach. The first explicit online nonparametric algorithms for regression with minimax rates were obtained in [4].

The nonparametric online algorithm of [6] is known to have a suboptimal regret bound for Lipschitz classes of functions. However, it is a simple and efficient algorithm, well suited to the design of extensions that exhibit different forms of adaptivity to the data sequence. For example, the paper [9] derived a variant that automatically adapts to the intrinsic dimension of the data manifold. Our work explores an alternative direction of adaptivity, mainly aimed at taming the effect of the curse of dimensionality in nonparametric prediction through the learning of an appropriate Mahalanobis distance on the instance space. There is a rich literature on metric learning (see, e.g., the survey [2]) where the Mahalanobis metric $\|\cdot\|_M$ is typically learned through minimization of the *pairwise loss function* of the form $\ell(M, x, x')$. This loss is high whenever dissimilar pairs of $x$ and $x'$ are close in the Mahalanobis metric, and whenever similar ones are far apart in the same metric —see, e.g., [19]. The works [5, 7, 18] analyzed generalization and consistency properties of online learning algorithms employing pairwise losses.

In this work we are primarily interested in using a metric $\|\cdot\|_M$ where $M$ is close to the gradient outer product matrix of the best model in the reference class of functions. As we are not aware whether pairwise loss functions can indeed consistently recover such metrics, we directly estimate the gradient outer product matrix. This approach to metric learning was mostly explored in statistics —e.g., by locally-linear Empirical Risk Minimization on RKHS [12, 11], and through Stochastic Gradient Descent [3]. Our learning approach combines —in a phased manner— a Mahalanobis metric extension of the algorithm by [6] with the estimator of [14]. Our work is also similar in spirit to the "gradient weights" approach of [8], which learns a distance based on a simpler diagonal matrix.

**Preliminaries and notation.** Let $\mathcal{B}(z, r) \subset \mathbb{R}^d$ be the ball of center $z$ and radius $r > 0$ and let $\mathcal{B}(r) = \mathcal{B}(0, r)$. We assume instances $x$ belong to $\mathcal{X} \equiv \mathcal{B}(1)$ and labels $y$ belong to $\mathcal{Y} \equiv [0, 1]$.

We consider the following online learning protocol with oblivious adversary. Given an unknown sequence $(x_1, y_1), (x_2, y_2), \cdots \in \mathcal{X} \times \mathcal{Y}$ of instances and labels, for every round $t = 1, 2, \ldots$

1. the environment reveals instance $x_t \in \mathcal{X}$;
2. the learner selects an action $\widehat{y}_t \in \mathcal{Y}$ and incurs the square loss $\ell_t(\widehat{y}_t) = (\widehat{y}_t - y_t)^2$;
3. the learner observes $y_t$.

Given a positive definite $d \times d$ matrix $M$, the norm $\|x - z\|_M$ induced by $M$ (a.k.a. Mahalanobis distance) is defined by $\sqrt{(x - z)^\top M (x - z)}$.

**Definition 1** (Covering and Packing Numbers). *An $\varepsilon$-cover of a set $S$ w.r.t. some metric $\rho$ is a set $\{x'_1, \ldots, x'_n\} \subseteq S$ such that for each $x \in S$ there exists $i \in \{1, \ldots, n\}$ such that $\rho(x, x'_i) \leq \varepsilon$. The covering number $\mathcal{N}(S, \varepsilon, \rho)$ is the smallest cardinality of a $\varepsilon$-cover.*

An $\varepsilon$-packing of a set $S$ w.r.t. some metric $\rho$ is a set $\{\boldsymbol{x}'_1, \ldots, \boldsymbol{x}'_m\} \subseteq S$ such that for any distinct $i, j \in \{1, \ldots, m\}$, we have $\rho(\boldsymbol{x}'_i, \boldsymbol{x}'_j) > \varepsilon$. The packing number $\mathcal{M}(S, \varepsilon, \rho)$ is the largest cardinality of a $\varepsilon$-packing.

It is well known that $\mathcal{M}(S, 2\varepsilon, \rho) \leq \mathcal{N}(S, \varepsilon, \rho) \leq \mathcal{M}(S, \varepsilon, \rho)$. For all differentiable $f : \mathcal{X} \to \mathcal{Y}$ and for any orthonormal basis $V \equiv \{\boldsymbol{u}_1, \ldots, \boldsymbol{u}_k\}$ with $k \leq d$ we define

$$\|\nabla_V f\|_\infty = \max_{\substack{\boldsymbol{v} \,\in\, \mathrm{span}(V) \\ \|\boldsymbol{v}\| = 1}} \sup_{\boldsymbol{x} \in \mathcal{X}} \nabla f(\boldsymbol{x})^\top \boldsymbol{v} \,.$$

If $V = \{\boldsymbol{u}\}$ we simply write $\|\nabla_{\boldsymbol{u}} f\|_\infty$.

In the following, $\boldsymbol{M}$ is a positive definite $d \times d$ matrix with eigenvalues $\lambda_1 \geq \cdots \geq \lambda_d > 0$ and eigenvectors $\boldsymbol{u}_1, \ldots, \boldsymbol{u}_d$. For each $k = 1, \ldots, d$ the truncated determinant is $\det_k(\boldsymbol{M}) = \lambda_1 \times \cdots \times \lambda_k$. The kappa function for the matrix $\boldsymbol{M}$ is defined by

$$\kappa(r, t) = \max\left\{ m \,:\, \lambda_m \geq t^{-\frac{2}{1+r}}, \, m = 1, \ldots, d \right\} \quad (5)$$

$$\text{for } t \geq 1 \text{ and } r = 1, \ldots, d.$$

Note that $\kappa(r+1, t) \leq \kappa(r, t)$. Now define the effective rank of $\boldsymbol{M}$ at horizon $t$ by

$$\rho_t = \min\left\{ r \,:\, \kappa(r, t) \leq r, \, r = 1, \ldots, d \right\} \,. \quad (6)$$

Since $\kappa(d, t) \leq d$ for all $t \geq 1$, this is a well defined quantity. Note that $\rho_1 \leq \rho_2 \leq \cdots \leq d$. Also, $\rho_t = d$ for all $t \geq 1$ when $\boldsymbol{M}$ is the $d \times d$ identity matrix. Note that the effective rank $\rho_t$ measures the number of eigenvalues that are larger than a threshold that shrinks with $t$. Hence matrices $\boldsymbol{M}$ with extremely light-tailed spectra cause $\rho_t$ to remain small even when $t$ grows large. This behaviour is shown in Figure 1.

Throughout the paper, we use $f \overset{\mathcal{O}}{=} (g)$ and $f \overset{\widetilde{\mathcal{O}}}{=} (g)$ to denote, respectively, $f = \mathcal{O}(g)$ and $f = \widetilde{\mathcal{O}}(g)$.

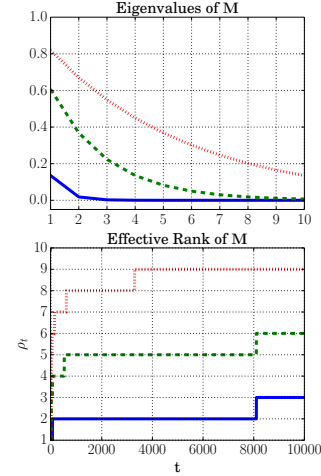

Figure 1: Quickly decreasing spectrum of $\boldsymbol{M}$ implies slow growth of its effective rank in $t$.

## 3 Online nonparametric learning with ellipsoid packing

In this section we present a variant (Algorithm 1) of the online nonparametric regression algorithm introduced in [6]. Since our analysis is invariant to rescalings of the matrix $\boldsymbol{M}$, without loss of generality we assume $\boldsymbol{M}$ has unit spectral radius (i.e., $\lambda_1 = 1$). Algorithm 1 sequentially constructs a packing of $\mathcal{X}$ using $\boldsymbol{M}$-ellipsoids centered on a subset of the past observed instances. At each step $t$, the label of the current instance $\boldsymbol{x}_t$ is predicted using the average $\widehat{y}_t$ of the labels of past instances that fell inside the ellipsoid whose center $\boldsymbol{x}_s$ is closest to $\boldsymbol{x}_t$ in the Mahalanobis metric. At the end of the step, if $\boldsymbol{x}_t$ was outside of the closest ellipsoid, then a new ellipsoid is created with center $\boldsymbol{x}_t$. The radii $\varepsilon_t$ of all ellipsoids are shrunk at rate $t^{-1/(1+\rho_t)}$. Note that efficient (i.e., logarithmic in the number of centers) implementations of approximate nearest-neighbor search for the active center $\boldsymbol{x}_s$ exist [10].

The core idea of the proof (deferred to the supplementary material) is to maintain a trade-off between the regret contribution of the ellipsoids and an additional regret term due to the approximation of $f$ by the Voronoi partitioning. The regret contribution of each ellipsoid is logarithmic in the number of predictions made. Since each instance is predicted by a single ellipsoid, if we ignore log factors the overall regret contribution is equal to the number of ellipsoids, which is essentially controlled by the packing number w.r.t. the metric defined by $\boldsymbol{M}$. The second regret term is due to the fact that —at any point of time— the prediction of the algorithm is constant within the Voronoi cells of $\mathcal{X}$ induced by the current centers (recall that we predict with nearest neighbor). Hence, we pay an extra term equal to the radius of the ellipsoids times the Lipschitz constant which depends on the directional Lipschitzness of $f$ with respect to the eigenbasis of $\boldsymbol{M}$.

**Theorem 1** (Regret with Fixed Metric). *Suppose Algorithm 1 is run with a positive definite matrix $\boldsymbol{M}$ with eigenbasis $\boldsymbol{u}_1, \ldots, \boldsymbol{u}_d$ and eigenvalues $1 = \lambda_1 \geq \cdots \geq \lambda_d > 0$. Then, for any differentiable*

---

**Algorithm 1** Nonparametric online regression

---

**Input:** Positive definite $d \times d$ matrix $\boldsymbol{M}$.

1: $S \leftarrow \varnothing$             ▷ Centers
2: **for** $t = 1, 2, \ldots$ **do**
3:     $\varepsilon_t \leftarrow t^{-\frac{1}{1+\rho_t}}$          ▷ Update radius
4:     Observe $\boldsymbol{x}_t$
5:     **if** $S \equiv \varnothing$ **then**
6:        $S \leftarrow \{t\}, \quad T_t \leftarrow \varnothing$          ▷ Create initial ball
7:     **end if**
8:     $s \leftarrow \arg\min\limits_{s \in S} \|\boldsymbol{x}_t - \boldsymbol{x}_s\|_{\boldsymbol{M}}$          ▷ Find active center
9:     **if** $T_s \equiv \varnothing$ **then**
10:      $y_t = \frac{1}{2}$
11:     **else**
12:      $\widehat{y}_t \leftarrow \dfrac{1}{|T_s|} \sum\limits_{t' \in T_s} y_{t'}$          ▷ Predict using active center
13:     **end if**
14:     Observe $y_t$
15:     **if** $\|\boldsymbol{x}_t - \boldsymbol{x}_s\|_{\boldsymbol{M}} \leq \varepsilon_t$ **then**
16:      $T_s \leftarrow T_s \cup \{t\}$          ▷ Update list for active center
17:     **else**
18:      $S \leftarrow S \cup \{s\}, \quad T_s \leftarrow \varnothing$          ▷ Create new center
19:     **end if**
20: **end for**

---

$f : \mathcal{X} \to \mathcal{Y}$ *we have that*

$$R_T(f) \stackrel{\widetilde{\mathcal{O}}}{=} \left( \sqrt{\det{}_\kappa(\boldsymbol{M})} + \sqrt{\sum_{i=1}^{d} \frac{\|\nabla_{\boldsymbol{u}_i} f\|_\infty^2}{\lambda_i}} \right) T^{\frac{\rho_T}{1+\rho_T}}$$

*where* $\kappa = \kappa(\rho_T, T) \leq \rho_T \leq d$.

We first prove two technical lemmas about packings of ellipsoids.

**Lemma 1** (Volumetric packing bound). *Consider a pair of norms* $\|\cdot\|, \|\cdot\|'$ *and let* $B, B' \subset \mathbb{R}^d$ *be the corresponding unit balls. Then*

$$\mathcal{M}(B, \varepsilon, \|\cdot\|') \leq \frac{\mathrm{vol}\left(B + \frac{\varepsilon}{2} B'\right)}{\mathrm{vol}\left(\frac{\varepsilon}{2} B'\right)} \ .$$

**Lemma 2** (Ellipsoid packing bound). *If* $B$ *is the unit Euclidean ball then*

$$\mathcal{M}\left(B, \varepsilon, \|\cdot\|_{\boldsymbol{M}}\right) \leq \left(\frac{8\sqrt{2}}{\varepsilon}\right)^s \prod_{i=1}^{s} \sqrt{\lambda_i} \qquad \text{where} \qquad s = \max\left\{i : \sqrt{\lambda_i} \geq \varepsilon, i = 1, \ldots, d\right\} \ .$$

The following lemma states that whenever $f$ has bounded partial derivatives with respect to the eigenbase of $\boldsymbol{M}$, then $f$ is Lipschitz with respect to $\|\cdot\|_{\boldsymbol{M}}$.

**Lemma 3** (Bounded derivatives imply Lipschitzness in $\boldsymbol{M}$-metric). *Let* $f : \mathcal{X} \to \mathbb{R}$ *be everywhere differentiable. Then for any* $\boldsymbol{x}, \boldsymbol{x}' \in \mathcal{X}$,

$$\left|f(\boldsymbol{x}) - f(\boldsymbol{x}')\right| \leq \|\boldsymbol{x} - \boldsymbol{x}'\|_{\boldsymbol{M}} \sqrt{\sum_{i=1}^{d} \frac{\|\nabla_{\boldsymbol{u}_i} f\|_\infty^2}{\lambda_i}} \ .$$

## 4   Learning while learning the metric

In this section, we assume instances $\boldsymbol{x}_t$ are realizations of i.i.d. random variables $\boldsymbol{X}_t$ drawn according to some fixed and unknown distribution $\mu$ which has a continuous density on its support $\mathcal{X}$. We also

assume labels $y_t$ are generated according to the noise model $y_t = f_0(\boldsymbol{x}_t) + \nu(\boldsymbol{x}_t)$, where $f_0$ is some unknown regression function and $\nu(\boldsymbol{x})$ is a subgaussian zero-mean random variable for all $\boldsymbol{x} \in \mathcal{X}$. We then simply write $R_T$ to denote the regret $R_T(f_0)$. Note that $R_T$ is now a random variable which we bound with high probability.

We now show how the nonparametric online learning algorithm (Algorithm 1) of Section 3 can be combined with an algorithm that learns an estimate

$$\widehat{\boldsymbol{G}}_n = \frac{1}{n}\sum_{t=1}^{n}\widehat{\nabla}f_0(\boldsymbol{x}_t)\widehat{\nabla}f_0(\boldsymbol{x}_t)^\top \tag{7}$$

of the expected outer product gradient matrix $\boldsymbol{G} = \mathbb{E}\left[\nabla f_0(\boldsymbol{X})\nabla f_0(\boldsymbol{X})^\top\right]$. The algorithm (described in the supplementary material) is consistent under the following assumptions. Let $\mathcal{X}(\tau)$ be $\mathcal{X}$ blown up by a factor of $1 + \tau$.

**Assumption 1.**

1. *There exists $\tau_0 > 0$ such that $f_0$ is continuously differentiable on $\mathcal{X}(\tau_0)$.*
2. *There exists $G > 0$ such that $\max\limits_{\boldsymbol{x}\in\mathcal{X}(\tau_0)} \|\nabla f_0(\boldsymbol{x})\| \leq G$.*
3. *The distribution $\mu$ is full-dimensional: there exists $C_\mu > 0$ such that for all $\boldsymbol{x} \in \mathcal{X}$ and $\varepsilon > 0$, $\mu\big(\mathcal{B}(\boldsymbol{x},\varepsilon)\big) \geq C_\mu \varepsilon^d$.*

In particular, the next lemma states that, under Assumption 1, $\widehat{\boldsymbol{G}}_n$ is a consistent estimate of $\boldsymbol{G}$.

**Lemma 4** ([14, Theorem 1]). *If Assumption 1 holds, then there exists a nonnegative and nonincreasing sequence $\{\gamma_n\}_{n\geq1}$ such that for all $n$, the estimated gradient outerproduct (7) computed with parameters $\varepsilon_n > 0$, and $0 < \tau_n < \tau_0$ satisfies $\left\|\widehat{\boldsymbol{G}}_n - \boldsymbol{G}\right\|_2 \leq \gamma_n$ with high probability with respect do the random draw of $\boldsymbol{X}_1,\ldots,\boldsymbol{X}_n$. Moreover, if $\tau_n = \Theta(\varepsilon_n^{1/4})$, $\varepsilon_n = \Omega\Big(\big(\ln n\big)^{\frac{2}{d}}n^{-\frac{1}{d}}\Big)$, and $\varepsilon_n = \mathcal{O}\Big(n^{-\frac{1}{2(d+1)}}\Big)$ then $\gamma_n \to 0$ as $n \to \infty$.*

Our algorithm works in phases $i = 1, 2, \ldots$ where phase $i$ has length $n(i) = 2^i$. Let $T(i) = 2^{i+1} - 2$ be the index of the last time step in phase $i$. The algorithm uses a nonincreasing regularization sequence $\overline{\gamma}_0 \geq \overline{\gamma}_1 \geq \cdots > 0$. Let $\widehat{\boldsymbol{M}}(0) = \overline{\gamma}_0\boldsymbol{I}$. During each phase $i$, the algorithm predicts the data points by running Algorithm 1 with $\boldsymbol{M} = \widehat{\boldsymbol{M}}(i-1)/\big\|\widehat{\boldsymbol{M}}(i-1)\big\|_2$ (where $\|\cdot\|_2$ denotes the spectral norm). Simultaneously, the gradient outer product estimate (7) is trained over the same data points. At the end of phase $i$, the current gradient outer product estimate $\widehat{\boldsymbol{G}}(i) = \widehat{\boldsymbol{G}}_{T(i)}$ is used to form a new matrix $\widehat{\boldsymbol{M}}(i) = \widehat{\boldsymbol{G}}(i) + \overline{\gamma}_{T(i)}\boldsymbol{I}$. Algorithm 1 is then restarted in phase $i + 1$ with $\boldsymbol{M} = \widehat{\boldsymbol{M}}(i)/\big\|\widehat{\boldsymbol{M}}(i)\big\|_2$. Note that the metric learning algorithm can be also implemented efficiently through nearest-neighbor search as explained in [14].

Let $\mu_1 \geq \mu_2 \geq \cdots \geq \mu_d$ be the eigenvalues and $\boldsymbol{u}_1,\ldots,\boldsymbol{u}_d$ be the eigenvectors of $\boldsymbol{G}$. We define the $j$-th eigenvalue separation $\Delta_j$ by

$$\Delta_j = \min_{k\neq j}\left|\mu_j - \mu_k\right|.$$

For any $\Delta > 0$ define also $V_\Delta \equiv \big\{\boldsymbol{u}_j : |\mu_j - \mu_k| \geq \Delta,\ k \neq j\big\}$ and $V_\Delta^\perp = \{\boldsymbol{u}_1,\ldots,\boldsymbol{u}_d\} \setminus V_\Delta$. Our results are expressed in terms of the effective rank (6) of $\boldsymbol{G}$ at horizon $T$. However, in order to account for the error in estimating the eigenvalues of $\boldsymbol{G}$, we define the effective rank $\widetilde{\rho}_t$ with respect to the following slight variant of the function kappa,

$$\widetilde{\kappa}(r,t) = \max\left\{m : \mu_m + 2\overline{\gamma}_t \geq \mu_1 t^{-\frac{2}{1+r}},\ m = 1,\ldots,d\right\} \quad t \geq 1 \text{ and } r = 1,\ldots,d.$$

Let $\widehat{\boldsymbol{M}}(i)$ be the estimated gradient outer product constructed at the end of phase $i$, and let $\widehat{\mu}_1(i) + \overline{\gamma}(i) \geq \cdots \geq \widehat{\mu}_d(i) + \overline{\gamma}(i)$ and $\widehat{\boldsymbol{u}}_1(i),\ldots,\widehat{\boldsymbol{u}}_d(i)$ be the eigenvalues and eigenvectors of $\widehat{\boldsymbol{M}}(i)$, where we also write $\overline{\gamma}(i)$ to denote $\overline{\gamma}_{T(i)}$. We use $\widehat{\kappa}$ to denote the kappa function with estimated eigenvalues and $\widehat{\rho}$ to denote the effective rank defined through $\widehat{\kappa}$. We start with a technical lemma.

**Lemma 5.** *Let $\mu_d, \alpha > 0$ and $d \geq 1$. Then the derivative of $F(t) = \big(\mu_d + 2\big(T_0 + t\big)^{-\alpha}\big)t^{\frac{2}{1+d}}$ is positive for all $t \geq 1$ when $T_0 \geq \left(\frac{d+1}{2\mu_d}\right)^{1/\alpha}$.*

*Proof.* We have that $F'(t) \geq 0$ if and only if $t \leq \frac{2(T_0+t)}{\alpha(d+1)}\left(1 + (T_0+t)^\alpha \mu_d\right)$. This is implied by

$$t \leq \frac{2\mu_d(T_0+t)^{1+\alpha}}{\alpha(d+1)} \quad \text{or, equivalently,} \quad T_0 \geq A^{1/(1+\alpha)}t^{1/(1+\alpha)} - t$$

where $A = \alpha(d+1)/(2\mu_d)$. The right-hand side $A^{1/(1+\alpha)}t^{1/(1+\alpha)} - t$ is a concave function of $t$. Hence the maximum is found at the value of $t$ where the derivative is zero, this value satisfies

$$\frac{A^{1/(1+\alpha)}}{1+\alpha}t^{-\alpha/(1+\alpha)} = 1 \quad \text{which solved for } t \text{ gives} \quad t = A^{1/\alpha}(1+\alpha)^{-(1+\alpha)/\alpha} .$$

Substituting this value of $t$ in $A^{1/(1+\alpha)}t^{1/(1+\alpha)} - t$ gives the condition $T_0 \geq A^{1/\alpha}\alpha(1+\alpha)^{-(1+\alpha)/\alpha}$
which is satisfied when $T_0 \geq \left(\frac{d+1}{2\mu_d}\right)^{1/\alpha}$. $\qquad\qquad\square$

**Theorem 2.** *Suppose Assumption 1 holds. If the algorithm is ran with a regularization sequence $\overline{\gamma}_0 = 1$ and $\overline{\gamma}_t = t^{-\alpha}$ for some $\alpha > 0$ such that $\overline{\gamma}_t \geq \gamma_t$ for all $t \geq \left(d + 1/2\mu_d\right)^{1/\alpha}$ and for $\gamma_1 \geq \gamma_2 \geq \cdots > 0$ satisfying Lemma 4, then for any given $\Delta > 0$*

$$R_T \stackrel{\widetilde{\mathcal{O}}}{=} \left(1 + \sqrt{\sum_{j=1}^{d} \frac{\left(\left\|\nabla_{\boldsymbol{u}_j}f_0\right\|_\infty + \left\|\nabla_{V_\Delta^\perp}f_0\right\|_\infty\right)^2}{\mu_j/\mu_1}}\right) T^{\frac{\tilde{\rho}_T}{1+\tilde{\rho}_T}}$$

*with high probability with respect to the random draw of $\boldsymbol{X}_1, \ldots, \boldsymbol{X}_T$.*

Note that the asymptotic notation is hiding terms that depend on $1/\Delta$, hence we can not zero out the term $\left\|\nabla_{V_\Delta^\perp}f_0\right\|_\infty$ in the bound by taking $\Delta$ arbitrarily small.

*Proof.* Pick the smallest $i_0$ such that

$$T(i_0) \geq \left(\frac{d+1}{2\mu_d}\right)^{1/\alpha} \tag{8}$$

(we need this condition in the proof). The total regret in phases $1, 2, \ldots, i_0$ is bounded by $\left(d + 1/2\mu_d\right)^{1/\alpha} = \mathcal{O}(1)$. Let the value $\widehat{\rho}_{T(i)}$ at the end of phase $i$ be denoted by $\widehat{\rho}(i)$. By Theorem 1, the regret $R_T(i+1)$ of Algorithm 1 in each phase $i + 1 > i_0$ is deterministically upper bounded by

$$R_T(i+1) \leq \left(8\ln\left(e2^{i+1}\right)\left(8\sqrt{2}\right)^{\widehat{\rho}(i+1)} + 4\sqrt{\sum_{j=1}^{d} \frac{\left\|\nabla_{\widehat{\boldsymbol{u}}_j(i)}f_0\right\|_\infty^2}{\lambda_j(i)/\lambda_1(i)}}\right) 2^{(i+1)\frac{\widehat{\rho}(i+1)}{1+\widehat{\rho}(i+1)}} \tag{9}$$

where $\lambda_j(i) = \widehat{\mu}_j(i) + \overline{\gamma}(i)$. Here we used the trivial upper bound $\det_\kappa\left(\widehat{\boldsymbol{M}}(i)/\|\widehat{\boldsymbol{M}}(i)\|_2\right) \leq 1$ for all $\kappa = 1, \ldots, d$. Now assume that $\widehat{\mu}_1(i) + \overline{\gamma}(i) \leq \left(\widehat{\mu}_m(i) + \overline{\gamma}(i)\right)t^{\frac{2}{1+r}}$ for some $m, r \in \{1, \ldots, d\}$ and for some $t$ in phase $i + 1$. Hence, using Lemma 4 and $\gamma_t \leq \overline{\gamma}_t$, we have that

$$\max_{j=1,\ldots,d}\left|\widehat{\mu}_j(i) - \mu_j\right| \leq \left\|\widehat{\boldsymbol{G}}(i) - \boldsymbol{G}\right\|_2 \leq \overline{\gamma}(i) \quad \text{with high probability.} \tag{10}$$

where the first inequality is straightforward. Hence we may write

$$\mu_1 \leq \mu_1 - \gamma(i) + \overline{\gamma}(i) \leq \widehat{\mu}_1(i) + \overline{\gamma}(i) \leq \left(\widehat{\mu}_m(i) + \overline{\gamma}(i)\right)t^{\frac{2}{1+r}}$$
$$\leq \left(\mu_m + \gamma(i) + \overline{\gamma}(i)\right)t^{\frac{2}{1+r}} \qquad \text{(using Lemma 4)}$$
$$\leq \left(\mu_m + 2\overline{\gamma}(i)\right)t^{\frac{2}{1+r}} .$$

Recall $\overline{\gamma}(i) = T(i)^{-\alpha}$. Using Lemma 5, we observe that the derivative of

$$F(t) = \left(\mu_m + 2\left(T(i) + t\right)^{-\alpha}\right)t^{\frac{2}{1+r}}$$

is positive for all $t \geq 1$ when

$$T(i) \geq \left(\frac{r+1}{2\mu_d}\right)^{1/\alpha} \geq \left(\frac{r+1}{2\mu_m}\right)^{1/\alpha}$$

which is guaranteed by our choice (8). Hence, $\big(\mu_m + 2\overline{\gamma}(i)\big)t^{\frac{2}{1+r}} \le \big(\mu_m + 2\overline{\gamma}_T\big)\big)T^{\frac{2}{1+r}}$ and so

$$\frac{\widehat{\mu}_1(i) + \overline{\gamma}(i)}{\widehat{\mu}_m(i) + \overline{\gamma}(i)} \le t^{\frac{2}{1+r}} \qquad \text{implies} \qquad \frac{\mu_1}{\mu_m + 2\overline{\gamma}_T} \le T^{\frac{2}{1+r}} \ .$$

Recalling the definitions of $\widetilde{\kappa}$ and $\widehat{\kappa}$, this in turn implies $\widehat{\kappa}(r, t) \le \widetilde{\kappa}(r, T)$, which also gives $\widehat{\rho}_t \le \widetilde{\rho}_T$ for all $t \le T$. Next, we bound the approximation error in each individual eigenvalue of $\boldsymbol{G}$. By (10) we obtain, for any phase $i$ and for any $j = 1, \dots, d$,

$$\mu_j + 2\overline{\gamma}(i) \ge \mu_j + \gamma(i) + \overline{\gamma}(i) \ge \widehat{\mu}_j(i) + \overline{\gamma}(i) \ge \mu_j - \gamma(i) + \overline{\gamma}(i) \ge \mu_j \ .$$

Hence, bound (9) implies

$$R_T(i+1) \le \left( 8\ln\big(e2^{i+1}\big)12^{\widetilde{\rho}_T} + 4\sqrt{\big(\mu_1 + 2\overline{\gamma}(i)\big)\sum_{j=1}^{d}\frac{\big\|\nabla_{\widehat{\boldsymbol{u}}_j}f_0\big\|_\infty^2}{\mu_j}} \right)2^{(i+1)\frac{\widetilde{\rho}_T}{1+\widetilde{\rho}_T}} \ . \qquad (11)$$

The error in approximating the eigenvectors of $\boldsymbol{G}$ is controlled via the following first-order eigenvector approximation result from matrix perturbation theory [20, equation (10.2)], for any vector $\boldsymbol{v}$ of constant norm,

$$\boldsymbol{v}^\top\big(\widehat{\boldsymbol{u}}_j(i) - \boldsymbol{u}_j\big) = \sum_{k \ne j}\frac{\boldsymbol{u}_k^\top\big(\widehat{\boldsymbol{M}}(i) - \boldsymbol{G}\big)\boldsymbol{u}_j}{\mu_j - \mu_k}\boldsymbol{v}^\top\boldsymbol{u}_k + o\Big(\big\|\widehat{\boldsymbol{M}}(i) - \boldsymbol{G}\big\|_2^2\Big)$$

$$\le \sum_{k \ne j}\frac{2\overline{\gamma}(i)}{\mu_j - \mu_k}\boldsymbol{v}^\top\boldsymbol{u}_k + o\big(\overline{\gamma}(i)^2\big) \qquad (12)$$

where we used $\boldsymbol{u}_k^\top\big(\widehat{\boldsymbol{M}}(i) - \boldsymbol{G}\big)\boldsymbol{u}_j \le \big\|\widehat{\boldsymbol{M}}(i) - \boldsymbol{G}\big\|_2 \le \overline{\gamma}(i) + \gamma(i) \le 2\overline{\gamma}(i)$. Then for all $j$ such that $\boldsymbol{u}_j \in V_\Delta$,

$$\nabla f_0(\boldsymbol{x})^\top\big(\widehat{\boldsymbol{u}}_j(i) - \boldsymbol{u}_j\big) = \sum_{k \ne j}\frac{2\overline{\gamma}(i)}{\mu_j - \mu_k}\nabla f_0(\boldsymbol{x})^\top\boldsymbol{u}_k + o\big(\overline{\gamma}(i)^2\big)$$

$$\le \frac{2\overline{\gamma}(i)}{\Delta}\sqrt{d}\,\|\nabla f_0(\boldsymbol{x})\|_2 + o\big(\overline{\gamma}(i)^2\big) \ .$$

Note that the coefficients

$$\alpha_k = \frac{\boldsymbol{u}_k^\top\big(\widehat{\boldsymbol{M}}(i) - \boldsymbol{G}\big)\boldsymbol{u}_j}{\mu_j - \mu_k} + o\big(\overline{\gamma}(i)^2\big) \qquad k \ne j$$

are a subset of coordinate values of vector $\widehat{\boldsymbol{u}}_j(i) - \boldsymbol{u}_j$ w.r.t. the orthonormal basis $\boldsymbol{u}_1, \dots, \boldsymbol{u}_d$. Then, by Parseval's identity,

$$4 \ge \|\widehat{\boldsymbol{u}}_j(i) - \boldsymbol{u}_j\|_2^2 \ge \sum_{k \ne j}\alpha_k^2 \ .$$

Therefore, it must be that

$$\max_{k \ne j}\left|\frac{\boldsymbol{u}_k^\top\big(\widehat{\boldsymbol{M}}(i) - \boldsymbol{G}\big)\boldsymbol{u}_j}{\mu_j - \mu_k}\right| \le 2 + o\big(\overline{\gamma}(i)^2\big) \ .$$

For any $j$ such that $\boldsymbol{u}_j \in V_\Delta^\perp$, since $\mu_j - \mu_k \ge \Delta$ for all $\boldsymbol{u}_k \in V_\Delta$, we may write

$$\nabla f_0(\boldsymbol{x})^\top\big(\widehat{\boldsymbol{u}}_j(i) - \boldsymbol{u}_j\big)$$

$$\le \frac{2\overline{\gamma}(i)}{\Delta}\sum_{\boldsymbol{u}_k \in V_\Delta}\nabla f_0(\boldsymbol{x})^\top\boldsymbol{u}_k + \Big(2 + o\big(\overline{\gamma}(i)^2\big)\Big)\sum_{\boldsymbol{u}_k \in V_\Delta^\perp}\nabla f_0(\boldsymbol{x})^\top\boldsymbol{u}_k + o\big(\overline{\gamma}(i)^2\big)$$

$$\le \frac{2\overline{\gamma}(i)}{\Delta}\sqrt{d}\,\|\boldsymbol{P}_{V_\Delta}\nabla f_0(\boldsymbol{x})\|_2 + \Big(2 + o\big(\overline{\gamma}(i)^2\big)\Big)\sqrt{d}\|\boldsymbol{P}_{V_\Delta^\perp}\nabla f_0(\boldsymbol{x})\|_2 + o\big(\overline{\gamma}(i)^2\big)$$

where $\boldsymbol{P}_{V_\Delta}$ and $\boldsymbol{P}_{V_\Delta^\perp}$ are the orthogonal projections onto, respectively, $V_\Delta$ and $V_\Delta^\perp$. Therefore, we have that

$$
\begin{aligned}
\left\|\nabla_{\widehat{\boldsymbol{u}}_j} f_0\right\|_\infty &= \sup_{\boldsymbol{x}\in\mathcal{X}} \nabla f_0(\boldsymbol{x})^\top \widehat{\boldsymbol{u}}_j(i) = \sup_{\boldsymbol{x}\in\mathcal{X}} \nabla f_0(\boldsymbol{x})^\top \big(\widehat{\boldsymbol{u}}_j(i) - \boldsymbol{u}_j + \boldsymbol{u}_j\big) \\
&\leq \sup_{\boldsymbol{x}\in\mathcal{X}} \nabla f_0(\boldsymbol{x})^\top \boldsymbol{u}_j + \sup_{\boldsymbol{x}\in\mathcal{X}} \nabla f_0(\boldsymbol{x})^\top \big(\widehat{\boldsymbol{u}}_j(i) - \boldsymbol{u}_j\big) \\
&\leq \left\|\nabla_{\boldsymbol{u}_j} f_0\right\|_\infty + \frac{2\overline{\gamma}(i)}{\Delta}\sqrt{d}\,\left\|\nabla_{V_\Delta} f_0\right\|_\infty + \Big(2 + o\big(\overline{\gamma}(i)^2\big)\Big)\sqrt{d}\left\|\nabla_{V_\Delta^\perp} f_0\right\|_\infty + o\big(\overline{\gamma}(i)^2\big) \quad (13)
\end{aligned}
$$

Letting $\alpha_\Delta(i) = \frac{2\overline{\gamma}(i)}{\Delta}\sqrt{d}\,\left\|\nabla_{V_\Delta} f_0\right\|_\infty + \Big(2 + o\big(\overline{\gamma}(i)^2\big)\Big)\sqrt{d}\left\|\nabla_{V_\Delta^\perp} f_0\right\|_\infty + o\big(\overline{\gamma}(i)^2\big)$ we can upper bound (11) as follows

$$
R_T(i+1) \leq \left(8\ln\big(e2^{i+1}\big)12^{\widetilde{\rho}_T} + 4\sqrt{\big(\mu_1 + 2\overline{\gamma}(i)\big)\sum_{j=1}^d \frac{\big(\left\|\nabla_{\boldsymbol{u}_j} f_0\right\|_\infty + \alpha_\Delta(i)\big)^2}{\mu_j}}\right) 2^{(i+1)\frac{\widetilde{\rho}_T}{1+\widetilde{\rho}_T}} \ .
$$

Recall that, due to (10), the above holds at the end of each phase $i+1$ with high probability. Now observe that $\overline{\gamma}(i) = \mathcal{O}\big(2^{-\alpha i}\big)$ and so $\alpha_\Delta(i) \overset{\mathcal{O}}{=} \big(\left\|\nabla_{V_\Delta} f_0\right\|_\infty/\Delta + \left\|\nabla_{V_\Delta^\perp} f_0\right\|_\infty\big)$. Hence, by summing over phases $i = 1,\ldots,\lceil \log_2 T\rceil$ and applying the union bound,

$$
\begin{aligned}
R_T &= \sum_{i=1}^{\lceil \log_2 T\rceil} R_T(i) \\
&\leq \left(8\ln\big(eT\big)12^d + 4\sqrt{\big(\mu_1 + 2\overline{\gamma}(i-1)\big)\sum_{j=1}^d \frac{\big(\left\|\nabla_{\boldsymbol{u}_j} f_0\right\|_\infty + \alpha_\Delta(i-1)\big)^2}{\mu_j}}\right) \left(2^{\frac{\widetilde{\rho}_T}{1+\widetilde{\rho}_T}}\right)^i \\
&\overset{\widetilde{\mathcal{O}}}{=} \left(1 + \sum_{j=1}^d \frac{\big(\left\|\nabla_{\boldsymbol{u}_j} f_0\right\|_\infty + \left\|\nabla_{V_\Delta^\perp} f_0\right\|_\infty\big)^2}{\mu_j/\mu_1}\right) T^{\frac{\widetilde{\rho}_T}{1+\widetilde{\rho}_T}}
\end{aligned}
$$

concluding the proof. $\qquad\square$

## 5 Conclusions and future work

We presented an efficient algorithm for online nonparametric regression which adapts to the directions along which the regression function $f_0$ is smoother. It does so by learning the Mahalanobis metric through the estimation of the gradient outer product matrix $\mathbb{E}[\nabla f_0(\boldsymbol{X})\nabla f_0(\boldsymbol{X})^\top]$. As a preliminary result, we analyzed the regret of a generalized version of the algorithm from [6], capturing situations where one competes against functions with directional Lipschitzness with respect to an arbitrary Mahalanobis metric. Our main result is then obtained through a phased algorithm that estimates the gradient outer product matrix while running online nonparametric regression on the same sequence. Both algorithms automatically adapt to the effective rank of the metric.

This work could be extended by investigating a variant of Algorithm 1 for classification, in which ball radii shrink at a nonuniform rate, depending on the mistakes accumulated within each ball rather than on time. This could lead to the ability of competing against functions $f$ that are only locally Lipschitz. In addition, it is conceivable that under appropriate assumptions, a fraction of the balls could stop shrinking at a certain point when no more mistakes are made. This might yield better asymptotic bounds than those implied by Theorem 1, because $\rho_T$ would never attain the ambient dimension $d$.

### Acknowledgments

Authors would like to thank Sébastien Gerchinovitz and Samory Kpotufe for useful discussions on this work. IK would like to thank Google for travel support. This work also was in parts funded by the European Research Council (ERC) under the European Union's Horizon 2020 research and innovation programme (grant agreement no 637076).

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
