[Supplementary Material]

# Supplementary Material for "Nonparametric Online Regression while Learning the Metric"

**Ilja Kuzborskij**
EPFL
Switzerland
ilja.kuzborskij@gmail.com

**Nicolò Cesa-Bianchi**
Dipartimento di Informatica
Università degli Studi di Milano
Milano 20135, Italy
nicolo.cesa-bianchi@unimi.it

## 1   Nonparametric gradient learning

In this section we describe a nonparametric gradient learning algorithm introduced in [2]. Throughout this section, we assume instances $\boldsymbol{x}_t$ are realizations of i.i.d. random variables $\boldsymbol{X}_t$ drawn according to some fixed and unknown distribution $\mu$ which has a continuous density on its support $\mathcal{X}$. Labels $y_t$ are generated according to the noise model $y_t = f(\boldsymbol{x}_t) + \nu(\boldsymbol{x}_t)$, where $\nu(\boldsymbol{x})$ is a subgaussian zero-mean random variable for all $\boldsymbol{x} \in \mathcal{X}$. The algorithm computes a sequence of estimates $\widehat{f}_1, \widehat{f}_2, \ldots$ of the regression function $f$ through kernel regression. Let $\mathcal{X}_n \equiv \{\boldsymbol{x}_1, \ldots, \boldsymbol{x}_n\} \subset \mathcal{X}$ be the data observed so far and let $y_1, \ldots, y_n$ their corresponding labels. Let $K : \mathbb{R}_+ \to \mathbb{R}_+$ be a nonincreasing kernel, strictly positive on $[0, 1)$, and such that $K(1) = 0$. Then the estimate at time $n$ is defined by

$$\widehat{f}_n(\boldsymbol{x}) = \sum_{t=1}^n y_t \, \omega_t(\boldsymbol{x}) \quad \text{where} \quad \omega_t(\boldsymbol{x}) = \begin{cases} \dfrac{K\left(\|\boldsymbol{x} - \boldsymbol{x}_t\| / \varepsilon_n\right)}{\sum_{s=1}^n K\left(\|\boldsymbol{x} - \boldsymbol{x}_s\| / \varepsilon_n\right)} & \text{if } \mathcal{B}(\boldsymbol{x}, \varepsilon_n) \cap \mathcal{X}_n \neq \emptyset, \\ 1/n & \text{otherwise} \end{cases}$$

where $\varepsilon_n > 0$ is the kernel scaling parameter. We then approximate the gradient of $\widehat{f}$ at any given point through the finite difference method

$$\Delta_i(\boldsymbol{x}) = \frac{1}{2\tau_n}\left(\widehat{f}(\boldsymbol{x} + \tau_n \boldsymbol{e}_i) - \widehat{f}(\boldsymbol{x} - \tau_n \boldsymbol{e}_i)\right) \qquad \text{for } i = 1, \ldots, d$$

where $\tau_n > 0$ is a parameter. Let further

$$A_i(\boldsymbol{x}) = \mathbb{I}\left\{\min_{b \in \{-\tau_n, \tau_n\}} \mu_n\left(\mathcal{B}(\boldsymbol{x} + b\boldsymbol{e}_i, \varepsilon/2)\right) \geq \frac{2d}{n}(\ln 2n)\right\} \qquad \text{for } i = 1, \ldots, d$$

where $\mu_n$ is the empirical distribution of $\mu$ after observing $\mathcal{X}_n$, and define the gradient estimate

$$\widehat{\nabla} f(\boldsymbol{x}_t) = \left(\Delta_1(\boldsymbol{x}_t) A_1(\boldsymbol{x}_t), \ldots, \Delta_d(\boldsymbol{x}_t) A_d(\boldsymbol{x}_t)\right).$$

The algorithm outputs at time $n$ the gradient outer product estimate

$$\widehat{\boldsymbol{G}}_n = \frac{1}{n} \sum_{t=1}^n \widehat{\nabla} f(\boldsymbol{x}_t) \widehat{\nabla} f(\boldsymbol{x}_t)^\top$$

Let $\boldsymbol{G} = \mathbb{E}\left[\nabla f(\boldsymbol{X}) \nabla f(\boldsymbol{X})^\top\right]$ be the expected gradient outer product, where $\boldsymbol{X}$ has law $\mu$. The next lemma states that, under Assumption 1, $\widehat{\boldsymbol{G}}_n$ is a consistent estimate of $\boldsymbol{G}$.

**Lemma** (Consistency of the Expected Gradient Outerproduct Estimator [2, Theorem 1]). *If Assumption 1 holds, then there exists a nonnegative and nonincreasing sequence $\{\gamma_n\}_{n \geq 1}$ such that for all $n$, the estimated gradient outerproduct (7) computed with parameters $\varepsilon_n > 0$, and $0 < \tau_n < \tau_0$ satisfies $\left\|\widehat{\boldsymbol{G}}_n - \boldsymbol{G}\right\|_2 \leq \gamma_n$ with high probability with respect do the random draw of $\boldsymbol{X}_1, \ldots, \boldsymbol{X}_n$. Moreover, if $\tau_n = \Theta\left(\varepsilon_n^{1/4}\right)$, $\varepsilon_n = \Omega\left((\ln n)^{\frac{2}{d}} n^{-\frac{1}{d}}\right)$, and $\varepsilon_n = \mathcal{O}\left(n^{-\frac{1}{2(d+1)}}\right)$ then $\gamma_n \to 0$ as $n \to \infty$.*

The actual rate of convergence depends, in a complicated way, on parameters related to the distribution $\mu$ and the regression function $f$. In our application of Lemma 4 we assume $\gamma_n \leq n^{-\alpha}$ for all $n$ large enough and for some $\alpha > 0$. Note also that the convergence of $\widehat{G}_n$ to $G$ holds in probability with respect to the random draw of $X_1, \ldots, X_n$. Hence there is a confidence parameter $\delta$ which is not shown here. However, the dependence of the convergence rate on $\frac{1}{\delta}$ is only polylogarithmic and therefore not problematic for our applications.

## 2 Proofs from Section 3

**Lemma** (Volumetric packing bound). *Consider a pair of norms $\|\cdot\|, \|\cdot\|'$ and let $B, B' \subset \mathbb{R}^d$ be the corresponding unit balls. Then*

$$\mathcal{M}(B, \varepsilon, \|\cdot\|') \leq \frac{\text{vol}\left(B + \frac{\varepsilon}{2}B'\right)}{\text{vol}\left(\frac{\varepsilon}{2}B'\right)} .$$

*Proof.* Let $\{x_1, \ldots, x_M\}$ be a maximal $\varepsilon$-packing of $B$ according to $\|\cdot\|'$. Since we have a packing, the $\|\cdot\|'$-balls of radius $\varepsilon/2$ and centers $x_1, \ldots, x_M$ are disjoint, and their union is contained in $B + \frac{\varepsilon}{2}B'$. Thus,

$$M\text{vol}\left(\frac{\varepsilon}{2}B'\right) \leq \text{vol}\left(B + \frac{\varepsilon}{2}B'\right)$$

which concludes the proof. $\qquad\square$

**Lemma** (Ellipsoid packing bound). *If $B$ is the unit Euclidean ball then*

$$\mathcal{M}(B, \varepsilon, \|\cdot\|_{\boldsymbol{M}}) \leq \left(\frac{8\sqrt{2}}{\varepsilon}\right)^s \prod_{i=1}^s \sqrt{\lambda_i} \qquad \text{where} \qquad s = \max\left\{i : \sqrt{\lambda_i} \geq \varepsilon, \ i = 1, \ldots, d\right\} .$$

*Proof.* The change of variable $x' = \boldsymbol{M}^{1/2}x$ implies $\|x\|_2 = \|x'\|_{\boldsymbol{M}^{-1}}$ and $\|x\|_{\boldsymbol{M}} = \|x'\|_2$. Therefore $\mathcal{M}(B, \varepsilon, \|\cdot\|_{\boldsymbol{M}}) = \mathcal{M}(E, \varepsilon, \|\cdot\|_2)$ where $E \equiv \left\{x \in \mathbb{R}^d : \|x\|_{\boldsymbol{M}^{-1}} \leq 1\right\}$ is the unit ball in the norm $\|\cdot\|_{\boldsymbol{M}^{-1}}$. Next, we write the coordinates $(x_1, \ldots, x_d)$ of any point $x \in \mathbb{R}^d$ using the orthonormal basis $u_1, \ldots, u_d$. Consider the truncated ellipsoid $\widetilde{E} \equiv \{x \in E : x_i = 0, \ i = s+1, \ldots, d\}$. By adapting an argument from [3], we prove that any $\varepsilon$-cover of $\widetilde{E}$ according to $\|\cdot\|_2$ is also a $(\varepsilon\sqrt{2})$-cover of $E$ according to the same norm. Indeed, let $\widetilde{S} \subset \widetilde{E}$ be a $\varepsilon$-cover of $\widetilde{E}$. Fix any $x \in E$ and let

$$\min_{\widetilde{\boldsymbol{x}} \in \widetilde{S}} \|x - \widetilde{x}\|_2^2 = \min_{\widetilde{\boldsymbol{x}} \in \widetilde{S}} \sum_{j=1}^s (x_j - \widetilde{x}_j)^2 + \sum_{j=s+1}^d x_j^2$$

$$\leq \varepsilon^2 + \sum_{j=s+1}^d x_j^2 \qquad\qquad (\text{since } \widetilde{S} \text{ is a } \varepsilon\text{-covering of } \widetilde{E})$$

$$\leq \varepsilon^2 + \lambda_{s+1} \sum_{j=s+1}^d \frac{x_j^2}{\lambda_j} \qquad (\text{since } \lambda_{s+1}/\lambda_j \geq 1 \text{ for } j = s+1, \ldots, d)$$

$$\leq 2\varepsilon^2$$

where the last inequality holds since $\lambda_{s+1} \leq \varepsilon^2$ and since $\|x\|_{\boldsymbol{M}^{-1}}^2 = \sum_{i=1}^d x_i^2/\lambda_i \leq 1$ for any $x \in E$, where $x_i = u_i^\top x$ for all $i = 1, \ldots, d$. Let $B' \subset \mathbb{R}^d$ be the unit Euclidean ball, and let $\widetilde{B}' \equiv \{x \in B' : x_i = 0, \ i = s+1, \ldots, d\}$ be its truncated version. Since $\lambda_i \geq \varepsilon^2$ for $i = 1, \ldots, s$ we have that for all $x \in \varepsilon\widetilde{B}'$, $x_1^2 + \cdots + x_s^2 \leq \varepsilon^2$ and so

$$\|x\|_{\boldsymbol{M}^{-1}}^2 = \sum_{i=1}^s \frac{x_i^2}{\lambda_i} \leq \sum_{i=1}^s \frac{\varepsilon^2}{\lambda_i} \leq 1 .$$

Therefore $\varepsilon \widetilde{B}' \subseteq \widetilde{E}$ which implies $\mathrm{vol}\big(\widetilde{E} + \frac{\varepsilon}{2}\widetilde{B}'\big) \le \mathrm{vol}\big(2\widetilde{E}\big)$.

$$
\begin{aligned}
\mathcal{M}\big(E, 2\varepsilon\sqrt{2}, \|\cdot\|_2\big) &\le \mathcal{N}\big(E, \varepsilon\sqrt{2}, \|\cdot\|_2\big) \\
&\le \mathcal{N}\big(\widetilde{E}, \varepsilon, \|\cdot\|_2\big) \\
&\le \mathcal{M}\big(\widetilde{E}, \varepsilon, \|\cdot\|_2\big) \\
&\le \frac{\mathrm{vol}\left(\widetilde{E} + \frac{\varepsilon}{2}\widetilde{B}'\right)}{\mathrm{vol}\left(\frac{\varepsilon}{2}\widetilde{B}'\right)} \qquad \text{(by Lemma 1)} \\
&\le \frac{\mathrm{vol}\big(2\widetilde{E}\big)}{\mathrm{vol}\left(\frac{\varepsilon}{2}\widetilde{B}'\right)} = \left(\frac{4}{\varepsilon}\right)^s \frac{\mathrm{vol}\big(\widetilde{E}\big)}{\mathrm{vol}\big(\widetilde{B}'\big)}
\end{aligned}
$$

Now, using the standard formula for the volume of an ellipsoid,

$$
\mathrm{vol}\big(\widetilde{E}\big) = \mathrm{vol}\big(\widetilde{B}'\big) \prod_{i=1}^{s} \sqrt{\lambda_i}\,.
$$

This concludes the proof. $\qquad\square$

The following lemma states that whenever $f$ has bounded partial derivatives with respect to the eigenbase of $M$, then $f$ is Lipschitz with respect to $\|\cdot\|_M$.

**Lemma** (Bounded derivatives imply Lipschitzness in $M$-metric). *Let $f : \mathcal{X} \to \mathbb{R}$ be everywhere differentiable. Then for any $x, x' \in \mathcal{X}$,*

$$
\big|f(\boldsymbol{x}) - f(\boldsymbol{x}')\big| \le \|\boldsymbol{x} - \boldsymbol{x}'\|_{\boldsymbol{M}} \sqrt{\sum_{i=1}^{d} \frac{\|\nabla_{\boldsymbol{u}_i} f\|_\infty^2}{\lambda_i}}\,.
$$

*Proof.* By the mean value theorem, there exists a $\boldsymbol{z}$ on the segment joining $\boldsymbol{x}$ and $\boldsymbol{y}$ such that $f(\boldsymbol{x}) - f(\boldsymbol{y}) = \nabla f(\boldsymbol{z})^\top (\boldsymbol{x} - \boldsymbol{y})$. Hence

$$
\begin{aligned}
f(\boldsymbol{x}) - f(\boldsymbol{y}) &= \nabla f(\boldsymbol{z})^\top (\boldsymbol{x} - \boldsymbol{y}) \\
&= \sum_{i=1}^{d} \nabla f(\boldsymbol{z})^\top \boldsymbol{u}_i \boldsymbol{u}_i^\top (\boldsymbol{x} - \boldsymbol{y}) \\
&\le \sum_{i=1}^{d} \left( \sup_{\boldsymbol{z}' \in \mathcal{X}} \nabla f(\boldsymbol{z}')^\top \boldsymbol{u}_i \right) \boldsymbol{u}_i^\top (\boldsymbol{x} - \boldsymbol{y}) \\
&= \sum_{i=1}^{d} \frac{\|\nabla_{\boldsymbol{u}_i} f\|_\infty}{\sqrt{\lambda_i}} \left( \sqrt{\lambda_i} \boldsymbol{u}_i^\top (\boldsymbol{x} - \boldsymbol{y}) \right) \\
&\le \sqrt{\sum_{i=1}^{d} \frac{\|\nabla_{\boldsymbol{u}_i} f\|_\infty^2}{\lambda_i}} \sqrt{\sum_{i=1}^{d} \lambda_i \big(\boldsymbol{u}_i^\top (\boldsymbol{x} - \boldsymbol{y})\big)^2} \quad \text{(by the Cauchy-Schwarz inequality)} \\
&= \|\boldsymbol{x} - \boldsymbol{y}\|_{\boldsymbol{M}} \sqrt{\sum_{i=1}^{d} \frac{\|\nabla_{\boldsymbol{u}_i} f\|_\infty^2}{\lambda_i}}\,.
\end{aligned}
$$

By symmetry, we can upper bound $f(\boldsymbol{y}) - f(\boldsymbol{x})$ with the same quantity. $\qquad\square$

Now we are ready to prove the regret bound.

**Theorem** (Regret with Fixed Metric). *Suppose Algorithm 1 is run with a positive definite matrix $\boldsymbol{M}$ with eigenbasis $\boldsymbol{u}_1, \ldots, \boldsymbol{u}_d$ and eigenvalues $1 = \lambda_1 \ge \cdots \ge \lambda_d > 0$. Then, for any differentiable $f : \mathcal{X} \to \mathcal{Y}$ we have that*

$$
R_T(f) \overset{\widetilde{\mathcal{O}}}{=} \left( \sqrt{\det_\kappa(\boldsymbol{M})} + \sqrt{\sum_{i=1}^{d} \frac{\|\nabla_{\boldsymbol{u}_i} f\|_\infty^2}{\lambda_i}} \right) T^{\frac{\rho_T}{1 + \rho_T}}
$$

*where $\kappa = \kappa(\rho_T, T) \leq \rho_T \leq d$.*

*Proof.* Let $S_t$ be the value of the variable $S$ at the end of time $t$. Hence $S_0 = \varnothing$. The functions $\pi_t : \mathcal{X} \to \{1, \dots, t\}$ for $t = 1, 2, \dots$ map each data point $\boldsymbol{x}$ to its closest (in norm $\|\cdot\|_{\boldsymbol{M}}$) center in $S_{t-1}$,

$$\pi_t(\boldsymbol{x}) = \begin{cases} \underset{s \in S_{t-1}}{\arg\min} \|\boldsymbol{x} - \boldsymbol{x}_s\|_{\boldsymbol{M}} & \text{if } S_{t-1} \not\equiv \varnothing \\ t & \text{otherwise.} \end{cases}$$

The set $T_s$ contain all data points $\boldsymbol{x}_t$ that at time $t$ belonged to the ball with center $\boldsymbol{x}_s$ and radius $\varepsilon_t$,

$$T_s \equiv \{t : \|\boldsymbol{x}_t - \boldsymbol{x}_s\|_{\boldsymbol{M}} \leq \varepsilon_t, \ t = s, \dots, T\} \ .$$

Finally, $y_s^\star$ is the best fixed prediction for all examples $(\boldsymbol{x}_t, y_t)$ such that $t \in T_s$,

$$y_s^\star = \underset{y \in \mathcal{Y}}{\arg\min} \sum_{t \in T_s} \ell_t(y) = \frac{1}{|T_s|} \sum_{t \in T_s} y_t \ . \tag{1}$$

We proceed by decomposing the regret into a local (estimation) and a global (approximation) term,

$$R_T(f) = \sum_{t=1}^{T} \left( \ell_t(\widehat{y}_t) - \ell_t\big(f(\boldsymbol{x}_t)\big) \right) = \sum_{t=1}^{T} \left( \ell_t(\widehat{y}_t) - \ell_t\big(y_{\pi_t(\boldsymbol{x}_t)}^\star\big) \right) + \sum_{t=1}^{T} \left( \ell_t\big(y_{\pi_t(\boldsymbol{x}_t)}^\star\big) - \ell_t\big(f(\boldsymbol{x}_t)\big) \right) .$$

The estimation term is bounded as

$$\sum_{t=1}^{T} \left( \ell_t(\widehat{y}_t) - \ell_t\big(y_{\pi_t(\boldsymbol{x}_t)}^\star\big) \right) = \sum_{s \in S_T} \sum_{t \in T_s} \left( \ell_t(\widehat{y}_t) - \ell_t(y_s^\star) \right) \leq 8 \sum_{s \in S_T} \ln(e|N_s|) \leq 8 \ln(eT)|S_T| \ .$$

The first inequality is a known bound on the regret under square loss [1, page 43]. We upper bound the size of the final packing $S_T$ using Lemma 2,

$$|S_T| \leq \mathcal{M}\big(B, \varepsilon_T, \|\cdot\|_{\boldsymbol{M}}\big) \leq \left( \frac{8\sqrt{2}}{\varepsilon_T} \right)^\kappa \prod_{i=1}^{\kappa} \sqrt{\lambda_i} \leq (8\sqrt{2})^\kappa \sqrt{\det_\kappa(\boldsymbol{M})} T^{\frac{\kappa}{1+\rho_T}}$$

where $\kappa = \kappa(\rho_T, T)$. Therefore, since $\rho_T \geq \kappa(\rho_T, T)$,

$$\sum_{t=1}^{T} \left( \ell_t(\widehat{y}_t) - \ell_t\big(y_{\pi_t(\boldsymbol{x}_t)}^\star\big) \right) \leq 8 \ln(eT)(8\sqrt{2})^{\rho_T} \sqrt{\det_\kappa(\boldsymbol{M})} T^{\frac{\rho_T}{1+\rho_T}} \ . \tag{2}$$

Next, we bound the approximation term. Using (1) we have

$$\sum_{t=1}^{T} \left( \ell_t\big(y_{\pi_t(\boldsymbol{x}_t)}^\star\big) - \ell_t\big(f(\boldsymbol{x}_t)\big) \right) \leq \sum_{t=1}^{T} \left( \ell_t\big(f(\boldsymbol{x}_{\pi_t(\boldsymbol{x}_t)})\big) - \ell_t\big(f(\boldsymbol{x}_t)\big) \right) .$$

Note that $\ell_t$ is 2-Lipschitz because $y_t, \widehat{y}_t \in [0, 1]$. Hence, using Lemma 3,

$$\ell_t\big(f(\boldsymbol{x}_{\pi_t(\boldsymbol{x}_t)})\big) - \ell_t\big(f(\boldsymbol{x}_t)\big) \leq 2\big|f(\boldsymbol{x}_{\pi_t(\boldsymbol{x}_t)}) - f(\boldsymbol{x}_t)\big|$$

$$\leq 2 \big\|\boldsymbol{x}_t - \boldsymbol{x}_{\pi_t(\boldsymbol{x}_t)}\big\|_{\boldsymbol{M}} \sqrt{\sum_{i=1}^{d} \frac{\|\nabla_{\boldsymbol{u}_i} f\|_\infty^2}{\lambda_i}}$$

$$\leq 2\varepsilon_t \sqrt{\sum_{i=1}^{d} \frac{\|\nabla_{\boldsymbol{u}_i} f\|_\infty^2}{\lambda_i}} \ .$$

Recalling $\varepsilon_t = t^{-\frac{1}{1+\rho_t}}$ where $\rho_t \leq \rho_{t+1}$, we write

$$\sum_{t=1}^{T} t^{-\frac{1}{1+\rho_t}} \leq \sum_{t=1}^{T} t^{-\frac{1}{1+\rho_T}} \leq \int_0^T \tau^{-\frac{1}{1+\rho_T}} \, d\tau = \left( 1 + \frac{1}{\rho_T} \right) T^{\frac{\rho_T}{1+\rho_T}} \leq 2 T^{\frac{\rho_T}{1+\rho_T}} \ .$$

Thus we may write

$$\sum_{t=1}^{T} \left( \ell_t\big(y_{\pi_t(\boldsymbol{x}_t)}^\star\big) - \ell_t\big(f(\boldsymbol{x}_t)\big) \right) \leq 4 \left( \sqrt{\sum_{i=1}^{d} \frac{\|\nabla_{\boldsymbol{u}_i} f\|_\infty^2}{\lambda_i}} \right) T^{\frac{\rho_T}{1+\rho_T}} \ .$$

The proof is concluded by combining the above with (2). $\qquad\qquad\square$