[Reviews · NeurIPS 2017]

Reviewer 1



This paper describes a novel algorithm for online nonparametric regression problem. It employs Mahalanobis metric to obtain a better distance measurement in the traditional online nonparametric regression framework. In terms of theoretical analysis, the proposed algorithm improves the regret bound and achieves a competitive result to the state-of-the-art. The theoretical proof is well organized and correct to the reviewer. However, the novelty of the proposed algorithm may be limited. The overall algorithm consists two separable components: online nonparametric regression with Mahalanobis metric, the estimation of gradient outer-product. The first component is a straightforward extension from [7], which proposed a general framework including all kinds of distance metric spaces. In Algorithm 1, it is obvious that the proposed work specializes the metric space as Mahalanobis distance and keep other steps unchanged. Therefore, the statement that the proposed algorithm is a generalized version of the algorithm from [7] in line 244 is improper. This extension is valuable only when the motivation of incorporating Mahalanobis distance is clearly introduced and convincing. Otherwise, this work mainly discusses a special case in the framework of [7]. On the other hand, the significance of the proposed method and the related regret bound should be discussed with more details. Because compared with the original method in [7], the improvement of regret bound depends on the specific problem. The second component is an application from [14] to estimate the gradient outer-product. The authors should discuss the increased computational complexity brought by this phase. Some minor issues: 1. How does one compute the active center efficiently in line 8 of Algorithm 1 2. $\tilde{\rho}$ in Theorem 2 is not defined. 3. What's the relationship between smoother f_0 and Mahalanobis metric? 4. There is no experimental evaluation to demonstrate the improvement of the proposed method in comparison with the original framework in terms of accuracy or convergence rate.

Reviewer 2



This paper presents a theoretical exploration on online regression with simultaneous metric learning. The framework is based on previous work, with modifications such as employing ellipsoid packing and Mahalanobis distance. The use of L_infty norms to cope with potential spectral leaking is also interesting. The authors were well aware of the limitation of their work, and pointed to a few meaningful directions for future work. It could be discussed how the modification to the regression algorithm in Algorithm 1 (with ellipsoids' radii shrinking over time) might affect the scalability of their findings. For instance, Line 12 uses a simple averaging step to calculate the prediction. Would it be possible or reasonable to use a weighted average, with the weighting decided by the distances? Also, is it possible to apply the scheme given at the end on *regression* problem as well, i.e., using accumulated prediction error within each ball to control the ball radius? These may not fall well within the scope of the current paper, but some relevant discussions may be helpful. A minor correction: Line 201, "... is ran" -> "... is run"

Reviewer 3



This paper studies the online nonparametric regression problem. The authors first extend the algorithm of Hazan and Megiddo to accommodate arbitrary pre-specified Mahalanobis distances (i.e. metrics induced by positive definite matrices), and they then study the stochastic setting and propose a method to approximate the norm induced by the expected gradient outer product using an estimator proposed by Trivedi et al, 2014 by applying the previous algorithm over epochs of doubling length, and using the empirical outer product over the previous epoch (plus a regularization term) as the metric for the current epoch. The authors present a guarantee that generalizes Hazan and Megiddo in the static metric setting, and a guarantee in the estimation setting that reflects the approximation error. The paper is well-written, clear, and presents both a natural and interesting extension of previous work in this setting. It is also a nice combination of previous work. Here are some questions and suggestions: 1) On lines 50-52, the authors discuss how the expected gradient outer product matrix is a natural choice for the metric in the stochastic setting. Is it possible to prove that this is actually the best matrix? 2) On line 120, \eps-covering and \eps-packing numbers should probably be defined precisely. 3) From a readability standpoint, the paper refers to Lemma 4 on line 203 without ever introducing it. In fact, it's not included in the paper at all. It would be good to fix this. 4) The effective rank expression is somewhat technical. Can the authors present some intuitive examples demonstrating situations where the effective rank leads to much better guarantees? 5) On lines 317-318, why is it reasonable to assume that \gamma_n \leq n^{-\alpha} for n large enough for Theorem 2? Can the authors at least provide some compelling examples?